# Identification of Inhibitors and Drug Targets for Human Adenovirus Infections

**DOI:** 10.3390/v14050959

**Published:** 2022-05-04

**Authors:** Minli Liu, Lefang Jiang, Weihua Cao, Jianguo Wu, Xulin Chen

**Affiliations:** 1State Key Laboratory of Virology, Wuhan Institute of Virology, Chinese Academy of Sciences, Wuhan 430071, China; liuminli.work@outlook.com; 2College of Life Sciences, University of the Chinese Academy of Sciences, Beijing 100049, China; 3Guangdong Provincial Key Laboratory of Virology, Institute of Medical Microbiology, Jinan University, Guangzhou 510632, China; lfjiang28@163.com (L.J.); caoweihua4u@163.com (W.C.)

**Keywords:** adenovirus, antiviral, drug screening, heat shock protein 90, mTOR signaling pathway, protein tyrosine kinases

## Abstract

Adenoviruses can cause infections in people of all ages at all seasons of the year. Adenovirus infections cause mild to severe illnesses. Children, immunocompromised patients, or those with existing respiratory or cardiac disease are at higher risk. Unfortunately, there are no commercial drugs or vaccines available on the market for adenovirus infections. Therefore, there is an urgent need to discover new antiviral drugs or drug targets for adenovirus infections. To identify potential antiviral agents for adenovirus infections, we screened a drug library containing 2138 compounds, most of which are drugs with known targets and past phase I clinical trials. On a cell-based assay, we identified 131 hits that inhibit adenoviruses type 3 and 5. A secondary screen confirmed the antiviral effects of 59 inhibitors that inhibit the replication of adenoviruses type 3 or 5. Most of the inhibitors target heat shock protein, protein tyrosine kinase, the mTOR signaling pathway, and other host factors, suggesting that these host factors may be essential for replicating adenoviruses. Through this study, the newly identified adenovirus inhibitors may provide a start point for developing new antiviral drugs to treat adenovirus infections. Further validation of the identified drug targets can help the development of new therapeutics against adenovirus infections.

## 1. Introduction

Adenoviruses (AdVs) are nonenveloped DNA viruses consisting of an icosahedral capsid and an inner nucleoprotein core containing a linear double-stranded DNA genome of ∼36 kbp. Human adenoviruses can cause infections in a population at any age, especially in young children and infants, causing many symptoms. Usually, upper respiratory, ocular, and enteric infections may cause only mild diseases. In contrast, lower respiratory infections, such as bronchitis, bronchiolitis, and pneumonia, can be severe and fatal [1,2]. Furthermore, an adenovirus infection can cause severe diseases in immunocompromised patients [3]. In addition, adenoviruses have been extensively studied and used as vectors for gene therapy and vaccine development. Therefore, potential adenovirus infections may cause severe diseases in patients receiving gene therapies or vaccinations [4,5,6,7]. Because of the broad spectrum and complexity of diseases associated with human adenovirus infections, and there are no FDA-approved antiviral drugs to treat the infections, it is urgently needed to develop new drugs for antiviral therapies.

Due to the cost and duration of developing and approving novel antivirals, repositioning existing drugs as therapeutics for adenovirus infections is an attractive strategy. In addition, since the drug has already undergone safety evaluation for its initial approval process, that alleviates the risk of failure [8]. The choice of types of adenoviruses for drug screening is based on several factors: The prevalence, the importance in the severity of adenovirus infections, the viruses mostly studied, and the availability in the lab. Both AdV3 and AdV5 are prevalent in children and can cause severe diseases, including pneumonia and meningoencephalitis [1,9]. These adenoviruses are also involved in severe and frequent complications in transplanted and immunocompromised patients [3,10]. To identify potential antiviral agents for adenovirus infections, we screened a drug library containing 2138 compounds, most of which are drugs with known targets and passed phase I clinical trials. Using a cell-based assay, through primary and secondary screens, we identified a panel of new inhibitors with potent antiviral activities against both AdV3 and AdV5 in vitro. Interestingly, the hit compounds target several host factors, suggesting that these host molecules and the pathways involved may be essential for replicating adenoviruses. The identified adenovirus inhibitors may provide a starting point for developing anti-adenovirus drugs through further study. Furthermore, the identified host factors may serve as antiviral drug targets for adenovirus infections.

## 2. Materials and Methods

### 2.1. Cell Lines and Virus Strains

Vero cells (ATCC CCL-81) were cultured in DMEM medium (Gibco) supplemented with 10% FBS (Gibco), 100 U/mL penicillin, and 100 U/mL streptomycin. All cells were maintained in a 5% CO_2_ incubator at 37 °C during passage and virus infection. Human adenoviruses type 3 and 5 were provided by the National Virus Resource Center (Wuhan, China). All experiments in this study were conducted in the biosafety level 2 laboratory.

### 2.2. Chemicals and Antibodies

The screen compound library was purchased from Selleck Chemicals (Shanghai, China). The library consisted of 2138 bioactive compounds with a >95% purity. The library compounds were stored in dimethyl sulfoxide (DMSO) at a ten mM stock concentration. Mouse monoclonal antibody against adenovirus virus hexon protein was obtained from Santa Cruz Biotechnology (SC-58085, 1:1000 diluted; Santa Cruz). DAPI was purchased from Solarbio (C0060, 1:10,000 diluted, Solarbio), and Goat FITC-conjugated anti-mouse secondary antibody was purchased from Abbkine (A22110, 1:1000 diluted; Abbkine).

### 2.3. Screening of the Drug Library for Adenovirus Inhibitors

In the primary screen, 2138 compounds from the inhibitor library were added individually to 384-well source plates (Labcyte, LP-0200). Subsequently, 100 nL of each inhibitor compound or DMSO were transferred from 384-well source plates to sterile, clear-bottom View Plate 96-well destination plates (PerkinElmer, 6007460) using an acoustic droplet ejection (ADE) system (Echo 550, Labcyte, CA, USA). Then, 100 μL Vero cells at a density of 2 × 10^4^ cells/mL in DMEM medium with 2% FBS, and 100 μL virus solution containing AdV3 at a multiplicity of infection (MOI) of 0.55 or AdV5 at an MOI of 1.1, were added to each well of 96-well destination plates, leading to a final concentration of 5 μM for each compound in the assay plate. Wells containing uninfected cells serve as the control. After incubating at 37 °C for 48 h, cell viability based on the drug-induced cytotoxicity effect, and virus inhibition rate based on the reduced virus-induced cytopathic effect (CPE), were determined arbitrarily based on the morphological observation under a microscope. Compounds that showed more than 70% inhibition and less than 10% cytotoxicity from the primary screen were defined as hits and were subjected to a secondary screen. The inhibition rates were graded from none to 100% with a 10% interval, estimated through morphological observation.

In the secondary screen, the dose-dependent effects of hit compounds on cytotoxicity and anti-AdV activity over a range of concentrations from 0.02 to 5 μM were determined using cell viability assay and Indirect Immunofluorescence Assay (IFA), as described in the following methods.

### 2.4. Cell Viability Assay

The cytotoxicity of cells treated with or without the hit compound for 48 h was determined following the manufacturer’s protocol of the CellTiter-Glo^®^ assay. After adding 100 μL CellTiter-Glo^®^ assay reagent (Promega) to each well, the plates were incubated at room temperature for 10 min to stabilize the luminescent signal. Fluorescence associated with cell viability was measured using a multi-label plate reader (Wallac Envision, PerkinElmer, MA, USA).

### 2.5. Anti-AdV Activity Assay

Vero cells were seeded in 96 well plates at a density of 1 × 10^4^ cells/well and cultured to 80% confluency. The supernatant medium was discarded, and the cells were washed twice with PBS. The cells were inoculated with AdV3 or AdV5 virus in the presence of hit compound at the experimental concentrations. The cells were cultured in a 5% CO_2_ incubator at 37 °C for 48 h. Next, IFA was conducted to measure the anti-AdV activity based on the hexon level that correlated to adenovirus replication. The anti-AdV assay was used to confirm the concentration-dependent antiviral effect of all hit compounds obtained in the primary screen.

### 2.6. Indirect Immunofluorescence Assay (IFA) 

Cells were washed twice with PBS and fixed in PFA (4% paraformaldehyde in PBS) for 20 min at room temperature (RT). The fixed samples were washed three times with PBST (0.05% tween 20 in PBS), then incubated in blocking buffer (3% BSA, 0.3% Triton X-100 and 10% FBS in PBS) for 1 h at RT, and then in binding buffer (3% BSA, 0.3% Triton X-100 in PBS) with mouse monoclonal antibodies against adenovirus virus hexon protein (dilution 1:50) overnight at 4 °C. After three more rinses, cells were incubated in binding buffer with Goat FITC-conjugated anti-mouse secondary antibody (dilution 1:1000) and DAPI (dilution 1:10,000) for 1 h in darkness at RT. After three final rinses with PBST, the samples were observed using the Operetta^®^ CLS™ high-content analysis system (PerkinElmer, Waltham, MA, USA), and then the images were taken and analyzed.

### 2.7. Image Analysis

The fluorescent signals from the stained samples were analyzed with the Operetta^®^ CLS™ high-content analysis system (PerkinElmer, USA). The percentage of the infected cells in each well of the 96 well plate was automatically obtained and analyzed from 9 fields per well using Harmony high-content imaging and analysis software (PerkinElmer, USA) to observe enough cells for the reliability of the statistical data while maintaining a fast-scanning speed.

Cells were defined by DAPI staining, and FITC intensity represents viral replication level was measured in the defined region of interest (ROI). The background fluorescence value was measured in uninfected control cells. Cells with FITC intensity more than three times the control cells were defined as AdV replication positive cells. The percentages of AdV replication positive cells/total cells ratios were calculated.

### 2.8. Statistical Analyses

Data in the primary screen (Figure 1) were presented as mean ± SEM for each category of compounds. Data in the secondary screen were obtained according to the dose-response curves fitted using nonlinear regression, log[drug] vs response; variable slope (four parameters), in GraphPad Prism v8.0.0 (GraphPad Software, San Diego, CA, USA). The concentrations required to inhibit cytopathicity by 50% (EC_50_), reduce cell viability by 50% (CC_50_), and selective indices (SIs, which is equal to CC_50_/EC_50_) of compounds were calculated and presented in Table 1, Table 2, Table 3 and Table 4.

## 3. Results

### 3.1. High-Throughput Screening and Identification of Human Adenovirus Inhibitors

To screen inhibitors and potential drug targets for adenovirus virus infection, we ordered a customized library of 2138 inhibitors from Selleck Chemicals, with more than 200 targets relating to angiogenesis, autophagy, cell cycle, cytoskeletal signaling, apoptosis, DNA damage, endocrinology and hormones, epigenetics, G proteins and G-protein-coupled receptors (GPCRs), immunology and inflammation, mitogen-activated protein kinase (MAPK), microbiology, metabolism, neuronal signaling, nuclear factor-kappa B (NF-κB), proteases, protein tyrosine kinases, stem cells and Wnt, TGF-β/Smad, transmembrane transporters, and ubiquitin. Most compounds are drugs with known targets and past phase I clinical trials. We screened the library using a cell-based assay, in which Vero cells at the density of 2 × 10^4^ cells/well were infected with AdV3 and AdV5 at an MOI of 0.56 and 1.1, respectively. The adenovirus infection causes a severe cytopathogenic effect (CPE), which can be observed and scaled under a microscope or quantitively determined by a cell viability assay. The virus replication levels are correlated to the CPE. The reduction of CPE suggests the inhibition of virus replication mediated by inhibitor treatment.

In the primary screening, a total of 131 hits were found to reduce the CPE caused by AdV3 or AdV5 infection by at least 70% at five µM concentration compared to the untreated control and with no apparent cytotoxicity (with more than 90% cell viability) (Figure 1A,B). Those compounds with apparent cytotoxicity (less than 90% cell viability) were excluded from the hits since cytotoxicity itself can inhibit virus replication. The average inhibition rate of all compounds in the cytoskeletal signaling group is the highest, followed by that of the PI3K/Akt/mTOR group and MAPK group. One hundred and twelve hits and 90 hits inhibit AdV3 and AdV5 infection, respectively, and 71 hits inhibit both viruses.

To further confirm the inhibitory effect of hit compounds, we evaluated the dose-dependent antiviral effects of all hit compounds in the Vero cell-based anti-AdV activity assay against AdV3 or AdV5 infection in vitro. The concentrations of each hit compound required to reduce virus replication by 50% and reduce cell viability by 50% (EC_50_ and CC_50_, respectively) were calculated using Prism v.8 software (GraphPad Software, San Diego, CA). Considering that we only tested concentrations up to 5 µM for all the hit compounds, at which most of the compounds are not toxic to Vero cells. Therefore, most of the CC50s of hit compounds are more than five µM. We define a hit as a compound with a selective index of more than three on the secondary screen. As shown in Figure 2A,B, 59 hits were confirmed with antiviral activity against at least one of AdV3 and AdV5. Interestingly, we found that the targets of the hit compounds are mainly heat shock proteins (HSPs) of the cytoskeletal signaling (Figure 3A), mammalian target of rapamycin (mTOR) (Figure 3B), protein tyrosine kinases (PTKs) (Figure 3C), and a few other host factors (Figure 3D). These observations suggest that these host factors and their signaling pathways may be functionally related to adenovirus replication. 

### 3.2. Antiviral Activities of Hit Compounds Targeting HSP

HSPs constitute a significant group of molecular chaperones that assist in properly folding partially folded or denatured proteins, organizing correct protein conformation and preventing irreversible aggregation of damaged proteins [11,12,13]. Due to the broad range of functions of HSPs, their dysfunction causes many severe disorders to host cells and virus life cycle as well. Six major families of HSPs viz., HSP20, HSP40, HSP60, HSP70, HSP90, and HSP100 have been reported based on molecular weight and functions [14,15]. Hsp90 is required for the activation/maturation of more than 300 client protein substrates. Moreover, hsp90-dependent substrates are associated with all ten hallmarks of cancer, making Hsp90 an attractive target for developing cancer chemotherapeutics [16]. For this reason, most of the HSP inhibitors in the library target Hsp90.

Among the 16 hit compounds targeting Cytoskeletal Signaling, only PF-3758309 targets PAK4, which is involved in cytoskeletal organization, cellular morphogenesis, and survival [17,18]. The other 15 hits target HSP90 (Table 1). All fifteen hits inhibit AdV3, eleven hits inhibit AdV5, and eleven hits inhibit both viruses’ infection in Vero cells. Most of the HSP inhibitors inhibit adenovirus replication at nanomolar concentrations. Of note, due to the use of library compounds, only concentrations up to 5 µM were tested for cytotoxicity, and the endpoints of the CC_50_s of most of the hits are not determined. Therefore, many CC_50_s of hits are recorded as >5 µM. As shown in Table 1, most HSP inhibitors inhibit both AdV3 and AdV5 infection with similar EC_50_s. Overall, HSP inhibitors potently inhibit adenovirus replication. Only a few hit compounds, i.e., CH5138303, and 17-AAG, inhibit AdV3 more efficiently than AdV5, suggesting that different adenoviruses may involve different host factors in their replication.

### 3.3. Antiviral Activities of Hit Compounds Targeting mTOR

The mTOR is a 289-kDa serine/threonine protein kinase with multiple domains, which modulates metabolism, cellular survival, DNA replication, gene transcription, protein synthesis, and virus replication [19,20,21]. As shown in Table 2, amongst the 17 hit compounds, 13 of them target mTOR (Table 2). In addition, 12 and 8 hits in mTOR inhibit AdV3 and AdV5 infection in Vero cells, respectively. Seven hits inhibit both viruses. Most of the mTOR inhibitors inhibit adenovirus replication at nanomolar or sub-micromolar concentrations. Overall, mTOR inhibitors potently inhibit adenovirus replication in Vero cells. Most of the mTOR inhibitors inhibit both AdV3 and AdV5 infection with similar EC50s, suggesting that PI3K/Akt/mTOR pathway is critical for adenovirus replication. However, H 89 2HCl, cAMP-dependent protein kinase A (PKA) inhibitor, inhibits AdV3 much more efficiently, suggesting that PKA may be more critical in AdV3 replication than in AdV5 replication.

### 3.4. Antiviral Activities of Hit Compounds Targeting PTK

The protein tyrosine kinases (PTKs) are a large and diverse multigene family found only in Metazoans. Their principal functions involve regulating multicellular aspects of the organism. For example, cell to cell signals concerning growth, differentiation, adhesion, motility, and death, are transmitted through tyrosine kinases. The human genome, as currently sequenced, contains 90 tyrosine kinase genes and five presumed tyrosine kinase pseudogenes. Of the 90 tyrosine kinase genes, 58 are of the receptor type, as defined by encoding a protein with a predicted transmembrane domain. Based on the kinase domain sequence, these 58 receptor tyrosine kinases can be grouped into 20 subfamilies, including EGFR, FGFR, PDGFR, and VEGFR. The 32 non-receptor tyrosine kinases fall into ten subfamilies, including SRC and SYK [22,23,24].

Interestingly, the drug library screening identified eight receptor tyrosine kinase inhibitors, three target EGFR, one targets FGFR, one targets PDGFR, and three target VEGFR (Table 3). These receptor tyrosine kinase inhibitors inhibit adenovirus replication at nanomolar to micromolar concentrations. In addition, we identified three non-receptor tyrosine kinase inhibitors. They target SRC or SYK. A few PTK inhibitors do not exhibit an antiviral effect to AdV5 at a concentration up to 5 µM. However, they may inhibit AdV5 replication at higher concentrations. Nevertheless, many PTK inhibitors inhibit AdV3 more efficiently than AdV5, suggesting that virus-host interaction for AdV3 may be different from AdV5.

### 3.5. Antiviral Activities of Hit Compounds Targeting Other Host Factors

Interestingly, the screening of adenovirus inhibitors identified a panel of hit compounds that target various host factors other than HSPs, mTOR, and PTKs (Table 4), including Aurora kinase, CDK, CRM1, p38 MAPK, Raf, PKC, IMPOD, DHODH (Dehydrogenase). Among them, Aurora kinase, CDK, p38 MAPK, Raf, and PKC are all kinases. Their inhibitors usually target multiple kinase targets. For example, ENMD-2076 L-(+)-Tartaric acid, selective activity against Aurora A, 25-fold more selective for Aurora A than Aurora B and less potent to VEGFR2/KDR and VEGFR3, FGFR1 and FGFR2 and PDGFRα. The SI for ENMD-2076 L-(+)-Tartaric acid is around 3 for both AdV3 and AdV5, similar to the VEGFR inhibitors (Table 3). Whether the antiviral activity through an on-target (targeting aurora kinase) or an off-target (for example, targeting VEGFR) remains to be investigated [25]. AMG-900 is a potent and highly selective pan-Aurora kinases inhibitor for Aurora A/B/C. It is >10 fold selective for Aurora kinases than p38α, Tyk2, and JNK2. Phase 1 [26]. AMG 900 potently inhibits only AdV3 but not AdV5.

Chromosome Region Maintenance1 (CRM1), one of the Karyopherin family’s seven known nuclear export proteins, is the best characterized nuclear exporter. CRM1 is the sole nuclear exporter of several cellular growth and survival factors, including proteins and RNA. Chromosome Region Maintenance1 (CRM1) has a crucial role in viruses from diverse families, including retroviruses, orthomyxoviruses, paramyxoviruses, flaviviruses, coronaviruses, rhabdoviruses, herpesviruses, and adenoviruses (Appendix A) [27,28]. Two inhibitors targeting two dehydrogenases, inosine monophosphate dehydrogenase (IMPDH), and dihydroorotate dehydrogenase (DHODH), were identified to inhibit both AdV3 and AdV5. Inhibitors targeting IMPDH or DHODH were reported to inhibit a panel of viruses, including SARS-CoV-2, HIV-1, and HCV (Appendix A) [29,30]. Results from us and others suggest that CRM1, IMPDH and DHODH are attractive antiviral targets.

## 4. Discussion

Since there are no FDA-approved drugs available to treat adenovirus infections and current antiviral studies for adenoviruses are very limited, we conducted antiviral research with adenovirus infections using a drug repurposing strategy to identify potential antiviral drugs and drug targets by screening a customized drug library. The library consists of 2138 compounds, most of which are drugs with known targets and passed phase I clinical trials. The FDA-approved drugs are not included in this library because most drug targets are unknown since many were approved a long time ago, while the drug development technologies were much less advanced than they are today. Though the drugs in the library inhibit more than 200 targets relating to almost every aspect of the cellular process, the diversity of drug targets is still limited. For example, six major families of HSPs viz., HSP20, HSP40, HSP60, HSP70, HSP90, and HSP100 have been reported based on molecular weight and functions [14]. However, most of the HSP inhibitors in the library are HSP90 inhibitors because they are being developed for new anti-cancer therapeutics. There are very few inhibitors in the library targeting other HSPs. Therefore, the number of inhibitors for each host factor (target) is not even.

In the primary screen, 131 hits were identified by reduced CPE mediated by drug treatment that was observed under the microscope. To eliminate the bias in the primary screen, we determined the dose-dependent effects of hit compounds on cytotoxicity and anti-AdV activity in the secondary screen. Fifty-nine hits were confirmed with antiviral activity against at least one of AdV3 and AdV5. Most of the EC_50_s are at nanomolar or sub-micromolar concentrations. Considering that we only tested concentrations up to 5 µM for all the hit compounds, most compounds are not toxic to Vero cells. However, most of the CC50s of hit compounds are more than five µM. We define a hit as a compound with a selective index of more than three on the secondary screen. Therefore, the actual SIs for most of the inhibitors could be much higher than those listed in Table 1, Table 2, Table 3 and Table 4. This expectation is confirmed in the determination of the EC_50_s and CC_50_s of AUY922 and PU-H71, two HSP90 inhibitors, in Vero cells (data not shown). The selective indices for the two hit compounds were determined to be >751.3 and >1057 in Vero cells for AdV3 and AdV5, respectively. In addition, some of the hit compounds inhibit AdV3 more efficiently than AdV5. The reason could be that the replication of AdV3 may involve different host factors than that for AdV5. Alternatively, the differences in MOI in the anti-AdV assay and virus replication efficiency may contribute to the differences in EC50s.

For most of the compounds in the library, the targets are known. By analyzing the distribution of targets of the identified hit compounds (Figure 1, Figure 2 and Figure 3), we found that the hit compounds inhibit mainly heat shock proteins (HSPs) of the cytoskeletal signaling (Table 1), mammalian target of rapamycin (mTOR) (Table 2), protein tyrosine kinases (PTKs) (Table 3), and a few other host factors (Table 4). Like many other viruses, the replication of adenoviruses relies on various host machineries in DNA replication, transcription, and translation. Of the inhibitors we identified, only 17-AAG and KPT-335 were reported to inhibit the replication of adenovirus 5 [28,31]. However, many other inhibitors were reported to inhibit a panel of DNA and RNA viruses (Appendix A). For example, HSP90 inhibitor STA-9090 inhibits EBV, RUBV, HPV [32,33,34], and 17-AAG inhibits 16 different types of viruses, including Ebola viruses and coronaviruses (Appendix A) [35,36]. An mTOR inhibitor, Torin 2, can inhibit the replication of 10 viruses, including SARS-CoV-2 (Appendix A) [37,38]. The EGFR inhibitor DesMethyl Erlotinib inhibits ten viruses (Appendix A) [39,40]. Interestingly, the CRM1 inhibitor KPT-335 (Verdinexor) inhibits 11 viruses, including adenoviruses, human immunodeficiency viruses and influenza viruses [28] (Appendix A). The IMPDH inhibitor Mycophenolate Mofetil can inhibit the replication of 15 viruses, the DHODH inhibitor Vidofludimus can inhibit five viruses, both inhibitors can inhibit SARS-CoV-2, HIV-1, and HCV (Appendix A) [29,30]. These results suggest that CRM1, IMPDH, and DHODH are potential antiviral targets. For the convenience of references, we listed all the literature that reported any antiviral activities of the inhibitors we identified. Although the inhibitor’s mechanisms of action on adenovirus may not be the same as on the other viruses, the same host targets may be involved in the replication of adenoviruses.

Antiviral drugs inhibit viral replication by targeting viral components or host factors. Direct-acting agents may induce the production of mutant viral strains capable of overcoming the antiviral effects of drugs targeting viral components. On the other hand, drugs targeting host factors essential for the viral life cycle have a high drug-resistant barrier. Among the most targeted host factors, HSPs, mTOR, PTKs, and other host factors, HSP90 and CRM1 are broadly investigated in antiviral mechanisms. Many research works have shown that the molecular chaperone Hsp90 is universally required for viral protein homeostasis. As observed for endogenous cellular proteins, numerous viral proteins have been shown to require Hsp90 for their folding, assembly, and maturation. Therefore, HSP90 may serve as a novel drug target for developing broad-spectrum antiviral drugs [41]. It is not surprising that we identified 15 HSP90 inhibitors. Several HSP90 inhibitors were reported to inhibit the replication of adenoviruses by inhibiting the transcription of early and late genes of AdV, replication of viral DNA, and expression of viral proteins [31]. CRM1 is the best characterized nuclear exporter and the sole nuclear exporter of several cellular growth and survival factors, including proteins and RNA. CRM1 has a key role in viruses from diverse families. Interruption of CRM1-mediated export results in changes in viral replication, viral protein expression, incomplete viral assembly, reduced infectivity, and improved antiviral host immune responses [27]. Like other viruses, adenovirus replicate in cells and need the function of CRM1 both in nuclear import and export [42].

The data for identifying adenovirus inhibitors and new drug targets presented here are preliminary, and we, and possibly other researchers, need to conduct further studies. For example, the CC_50_s and EC_50_s of all hit compounds should be determined in different cells infected with multiple types of adenoviruses at different MOIs. In addition, the determination of each inhibitor’s antiviral mechanism of action, the validation of each drug target, and the evaluation of antiviral efficacy in vivo are needed. Nevertheless, of the 59 newly identified inhibitors, 57 drugs have never been reported to inhibit the replication of adenoviruses. Thirty-one drugs (5 targeting HSP90, 11 targeting PI3K/Akt/mTOR, 7 targeting protein tyrosine kinases, 8 targeting other host factors) have never been reported to inhibit any viruses. These newly identified adenovirus inhibitors may provide a good starting point for developing anti-adenovirus drugs through further study. Furthermore, the identified host targets may serve as antiviral drug targets for adenovirus infections by further validation study.

## Figures and Tables

**Figure 1 viruses-14-00959-f001:**
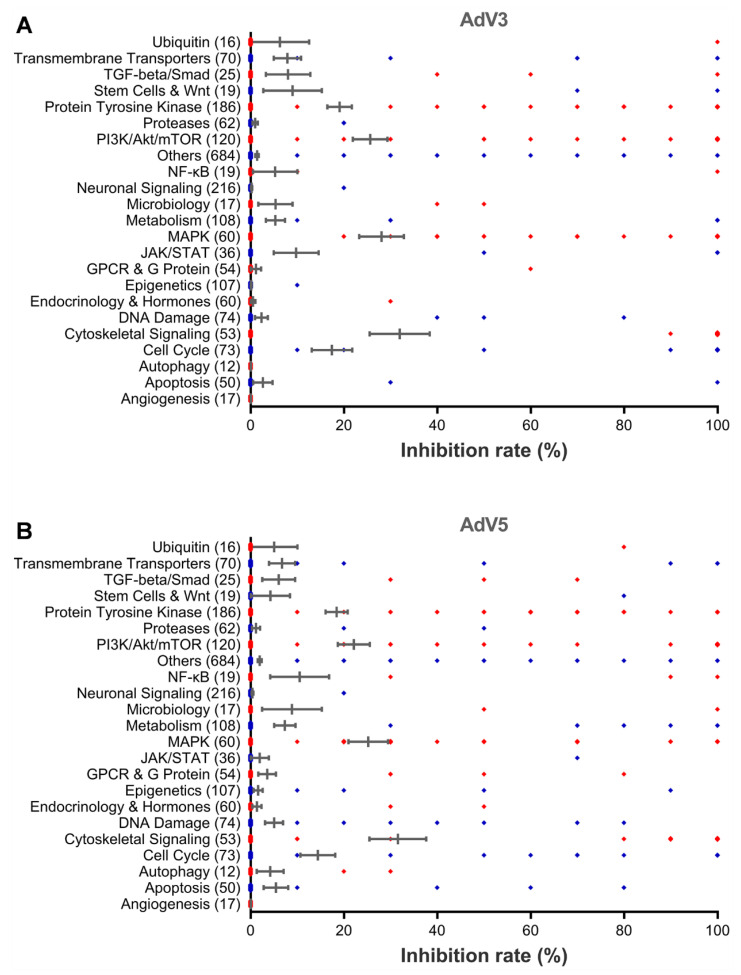
The cytopathic effect-based high-throughput screening identifies new antiviral agents against AdV replication. Vero cells were incubated with five μM of each library compound for 10 min and followed by the infection with the AdV3 at MOI 0.56 or AdV5 at MOI 1.1. After incubation at 37 °C for 48 hrs, cell viability and inhibition rates were determined. Graphs show the percentage of inhibition rates against AdV3 (**A**) or AdV5 (**B**) of all drugs in each category of drug target. The inhibition rate for each category is presented as mean ± SEM. The total compound number in each category is indicated in parentheses.

**Figure 2 viruses-14-00959-f002:**
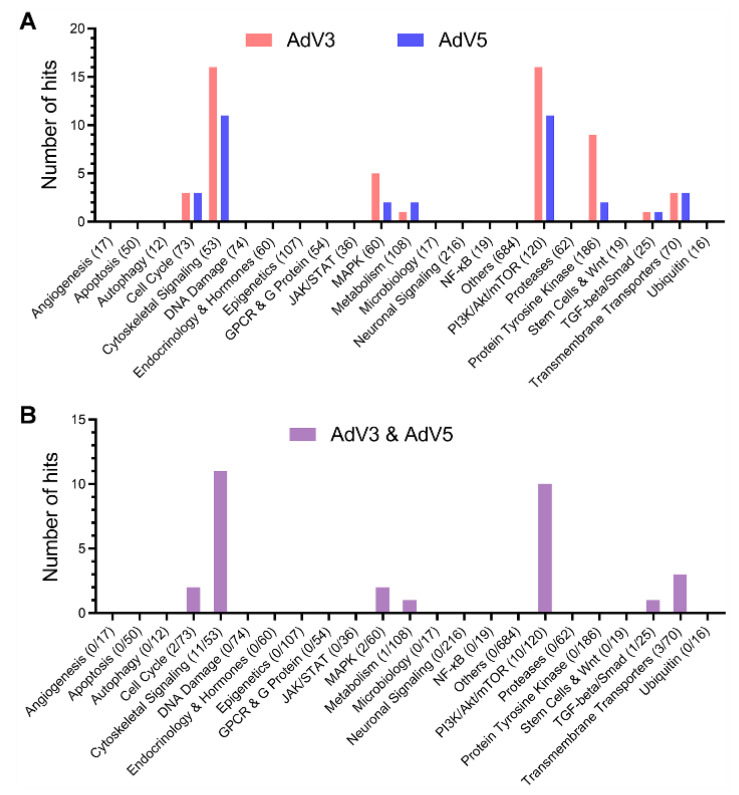
The secondary screening of the hits from the primary screen verifies the hits with dose-dependent antiviral activities. Vero cells were incubated with the serially diluted hit compounds from the primary screening for 10 min, followed by infection with AdV3 or AdV5 for 48 h. The viral replication levels were determined by IFA to detect hexon. The inhibition rates were calculated according to the methods in the anti-AdV activity assay. The cytotoxicity was measured by a cell viability assay using the CellTiter-Glo^®^ reagents over the serially diluted hit compounds. (**A**) The hit compound against AdV3 or AdV5 respectively categorizes the number of hits targeting each host factor or pathway in the secondary screening. (**B**) The number of hits inhibits both AdV3 and AdV5, targeting each host factor or pathway in the secondary screening. The ratio of hit number over the total compound number in each category is indicated in parentheses.

**Figure 3 viruses-14-00959-f003:**
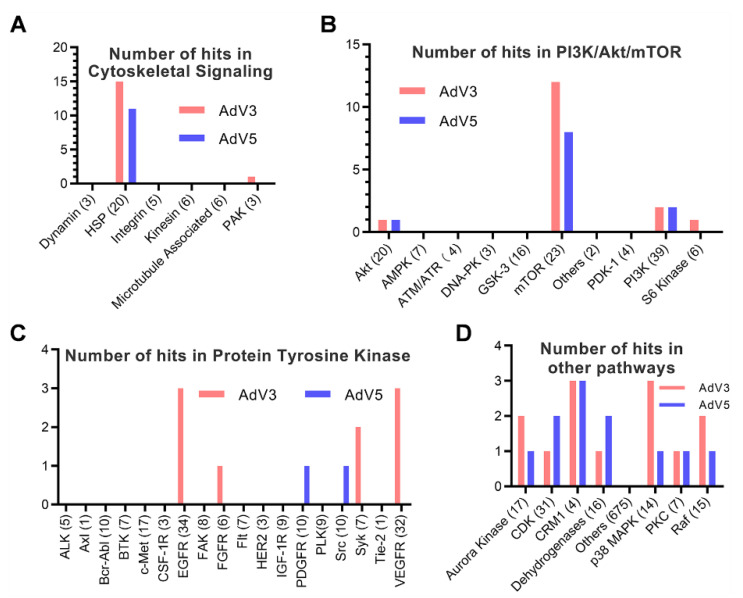
Analysis of the targets of hit compounds from the secondary screen. The target distribution analysis shows that drug targets of all hit compounds in the secondary screen can be categorized into four groups. (**A**) Cytoskeletal signaling. (**B**) PI3K/Akt/mTOR. (**C**) Protein Tyrosine Kinase, and (**D**) other pathways. The total compound number in each category is indicated in parentheses. The number of hits in each category is shown on the *Y*-axis.

**Table 1 viruses-14-00959-t001:** Anti-ADV activities of hit compounds targeting cytoskeletal signaling.

Target	Name	CC50 (µM)	EC50 (µM)		Selective Index	
			AdV3	AdV5	AdV3	AdV5
PAK ^1^	PF-3758309	>5.00	0.54	2.26	>9.29	>2.22
HSP90 ^2^	AUY922	>5.00	0.18	0.23	>28.56	>22.11
	17-AAG	>5.00	0.50	10.86	>9.93	>0.46
	17-DMAG	>5.00	0.22	0.43	>23.20	>11.58
	STA-9090	4.44	0.05	0.05	97.56	76.73
	AT13387	>5.00	0.08	0.18	>61.84	>27.13
	BIIB021	>5.00	0.10	0.24	>48.73	>21.2
	SNX-2112	>5.00	0.17	0.33	>28.87	>15.09
	PF-04929113	>5.00	0.19	0.40	>26.61	>12.46
	Geldanamycin	>5.00	1.48	>5.00	>3.38	ND ^3^
	HSP990	4.31	0.02	0.03	205.19	148.47
	XL888	>5.00	0.17	0.48	>29.69	>10.33
	NMS-E973	>5.00	0.92	4.94	>5.44	>1.01
	CH5138303	>5.00	0.09	0.65	>53.77	>7.72
	VER-49009	>5.00	0.03	0.08	>186.00	>60.00
	PU-H71	>5.00	0.74	2.20	>6.79	>2.27

^1^ PAK: p21-activated kinase; ^2^ HSP90: Heat shock protein 90. ^3^ ND:Not determined.

**Table 2 viruses-14-00959-t002:** Anti-ADV activities of hit compounds targeting PI3K/Akt/mTOR.

Target	Name	CC50 (µM)	EC50 (µM)		Selective Index	
			AdV3	AdV5	AdV3	AdV5
PKA ^1^	H 89 2HCl	>5.00	0.31	>5.00	>15.97	ND ^5^
PI3K ^2^	VS-5584	0.40	0.10	0.10	3.94	4.18
	GNE-317	>5.00	0.26	0.59	19.28	8.50
Akt ^3^	LY3023414	>5.00	0.90	1.31	>5.58	>3.83
mTOR ^4^	AZD8055	>5.00	0.11	0.16	>44.72	>31.87
	PP242	>5.00	1.03	1.97	>4.85	>2.54
	OSI-027	>5.00	1.38	2.40	>3.62	>2.08
	WYE-125132	>5.00	0.06	0.15	>89.17	>32.47
	WAY-600	3.11	0.56	1.09	5.51	2.86
	GDC-0980	3.00	0.08	0.27	36.44	10.96
	PF-04691502	>5.00	0.06	0.12	>86.01	>41.46
	AZD2014	4.05	0.02	0.07	171.31	56.82
	INK 128	0.85	0.01	0.01	161.78	61.27
	Torin 2	0.78	0.31	0.09	2.48	8.30
	Voxtalisib	2.26	0.47	2.72	4.83	0.83
	CC-223	>5.00	1.11	1.38	>4.50	>3.62
	GDC-0349	>5.00	1.38	2.62	>3.63	>1.91

^1^ PKA: (cAMP-dependent) protein kinase A; ^2^ PI3K: Phosphoinositide 3-kinase. ^3^ Akt: Protein kinase B; ^4^ mTOR: Mammalian target of rapamycin. ^5^ ND: Not determined.

**Table 3 viruses-14-00959-t003:** Anti-ADV activities of hit compounds targeting protein tyrosine kinase.

Target	Name	CC50 (µM)	EC50 (µM)		Selective Index	
			AdV3	AdV5	AdV3	AdV5
EGFR ^1^	AZD8931	>5.00	0.05	>5.00	>96.15	ND ^6^
	DesMethyl Erlotinib	>5.00	0.51	>5.00	>9.72	ND
	Dacomitinib	4.88	0.77	1.64	6.31	2.97
FGFR ^2^	AZD4547	51.45	5.86	ND	8.78	ND
PDGFR ^3^	Amuvatinib	8.34	3.97	1.80	2.10	4.64
VEGFR ^4^	Sorafenib Tosylate	5.93	1.27	3.87	4.67	1.53
	Apatinib	>5.00	0.32	3.60	>15.41	>1.39
	Regorafenib	4.06	0.44	1.50	9.30	2.72
SRC ^5^	KW-2449	>5.00	3.66	1.28	>1.37	>3.91
SYK ^6^	R406	>5.00	0.94	>5.00	>5.32	ND
	R788	1.56	0.23	1.06	6.65	1.48

^1^ EGFR: Epidermal growth factor receptor; ^2^ FGFR: Fibroblast growth factor receptor. ^3^ PDGFR: Platelet-derived growth factor receptor; ^4^ SRC: Short for sarcoma, a non-receptor tyrosine kinase. ^5^ SYK: Spleen tyrosine kinase, a non-receptor tyrosine kinase. ^6^ ND: Not determined.

**Table 4 viruses-14-00959-t004:** Anti-ADV activities of hit compounds targeting other host pathways.

Pathway	Target	Name	CC50 (µM)	EC50 (µM)		Selective Index	
				AdV3	AdV5	AdV3	AdV5
Cell Cycle	Aurora Kinase ^1^	ENMD-2076 L-(+)-Tartaric acid	4.34	1.35	1.39	3.22	3.14
		AMG-900	3.69	0.02	>5.00	226.12	ND ^9^
	CDK ^2^	BMS-265246	>5.00	0.64	0.6	>7.84	>8.35
		ON123300	>5.00	2.91	1.62	>1.72	>3.08
Transmembrane Transporters	CRM1 ^3^	KPT-276	>5.00	0.1	0.97	>50.46	>5.13
		KPT-330	1.58	0.1	0.52	15.45	3.04
		KPT-335	1.9	0.15	0.61	12.65	3.14
MAPK	p38 MAPK ^4^	Doramapimod	>5.00	0.35	>5	>14.24	ND
		PH-797804	>5.00	0.02	1.58	>250.00	>3.16
		Skepinone-L	2.14	0.18	2.56	11.84	0.83
	Raf ^5^	RAF265	>5.00	1.28	>5.00	>3.91	ND
		TAK-632	1.99	0.18	0.16	11.15	12.3
TGF-beta/Smad	PKC ^6^	Go6976	3.95	0.62	1.23	6.39	3.22
Metabolism	IMPDH ^7^	Mycophenolate Mofetil	>5.00	1.84	0.19	>2.72	>25.83
	DHODH ^8^	Vidofludimus	>5.00	1.63	0.94	>3.07	>5.35

^1^ Aurora Kinase: A serine/threonine kinase; ^2^ CDK: Cyclin-dependent kinase. ^3^ CRM1: Chromosomal Maintenance 1, also known as Exportin 1. ^4^ p38 MAPK: p38 mitogen-activated protein kinase; ^5^ Raf: rapid accelerated fibrosarcoma, a family of three serine/threonine-specific kinases (A-Raf, B-Raf, and Raf-1). ^6^ PKC: Potent protein kinase C. ^7^ IMPDH: Inosine monophosphate dehydrogenase. ^8^ DHODH: Dihydroorotate dehydrogenase; ^9^ ND: Not determined.

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
