# Peer review of "Identification of Inhibitors and Drug Targets for Human Adenovirus Infections"

_viruses, 2022, doi:10.3390/v14050959_

Round 1

Reviewer 1 Report

The work of Liu et al investigated a panel of chemical compounds for their antiviral activity against two human adenoviruses (HAdV), HAdV 3 from HAdV species B and HAdV 5 from species C. They first screened for the ability of drugs to reduce cytopathic effects and then determined concentrations inhibiting 50% of virus replication.

This works enables to identify cellular targets involved in Adenovirus replication.

Most targets identified have also been associated with the replication of other viruses.

The authors should address the following points to improve their manuscript.

Major points

Please explain more how the compound library was selected. Indeed, the composition highly influences the results. Whether some cellular protein families are not included, some potential targets won’t be identified.

Introduction.

Please explain the reasons to select Adv3 and Adv5.

Methods.

In methods the authors mention a volume used for the drugs. Is it correct that they used 100 nL (page 2 line 74) ?

Revise the sentence Line 104 to line 106.

2.5 How many dilutions were tested ? What was the dilution factor between successive dilutions  ?

Results.

The first sentence (line 140) is not clear.

Line 160. I do not understand the link between the sentence  lines 157-158 and the following sentence. Which compounds were excluded -à Flow chart ?

Figure 1. Mean +/- SEM are presented. What does the red and blue triangle mean ?

3.5. The results regarding the pathway MAPK are not presented in the text.

Discussion.

The authors explained that they did not test FDA approved drugs because drug targets are unknown. This reason is very surprising. As FDA approved drugs are already available on the market, they could be used more quickly. If some drugs were identified with interesting anti-adenovirus activity, their mechanisms could be investigated in further works.

The reason should be explained and discussed in more details.

The authors provided new data in the discussion. They should be replaced in the results section. They used another cell line (A549) not mentioned in methods. The authors must give more details and explain why they used another cell line.

Line 347. It is not sure that CRM1, IMPDH and DHOD are attractive targets. Because they seem to be involved in a large panel of viruses with different replication cycles, they might also be critical for cellular functions and inhibitors might significant side-effects in vivo. Please further discuss or revise this point.

Reviewer 2 Report

Manuscript by Liu et al., deals with finding of drugs inhibiting human adenovirus 3 and 5 (HAdV-3 & HAdV-5) growth. Based on their cell-based assay, the authors identified 131 hits that inhibiting HAdV-3 and -5. A secondary screen confirmed the antiviral effects of 59 inhibitors. Most of the inhibitors target HSPs and PTKs, suggesting that these host factors may be essential for replicating HAdVs. 

The manuscript is OK written, methods are described in details, experiments are more-or-less controlled. As such its is an interesting manuscript regarding the fact that there is an intensive hunt for anti-HAdV antivirals. It is mainly descriptive manuscript, without direct proof that the found hits really are working via the targeted proteins on HAdV replication. In other words, the manuscript lacks direct experimental evidence to support the findings, although I understand that it is difficult task to carry out in short time frame.Thats why the other adenovirologists, working on mechanisms, will find it an interesting manuscript and for sure we will use it as a source to study detailed mechanisms of HAdV replication. Thank you for this nice work!

Anyhow, I have the following comments:

L13-15: the first 2 sentences of the abstract are difficult to understand and grammatically wrong, rephrase them.

L15-16: this is wrong statement as there is a vaccine available, and in use, against HAdV-4 mainly for the US military recruits. So, make this statement correct.

L16: "urgent need "is a clear overstatement as most of us, we are not dying of HAdV infections. Use "need" only.

L51: why did the authors focus on HAdV-3 and -5? This has to be explained as well their potential connection to pathogenic/fatal infections and infections among immunocompromised/pediatric patients.

L148: why were the Vero cells chosen? Most of the HAdV studies are done using A549 cells. The reason has to be explained.

L152: "The virus replication levels are reversely correlated to the CPE. "This is not true, rephrase the sentence.

L151: it still remains unclear to me how the authors studies the CPE. By manual observation or using an automate method? 

L163: anti-AdV activity does not say too much. Add extra sentence(s), like  the reason for this assay, what is the exact read-out of it and how one should interpret it. Can be also wise to update the M&M about it.

Fig.1 and Fig.2: Was the drug added before infection (Fig1) and after infection (Fig2)? Or was the drug treatment done always as explained in Fig.1? The order of adding is important and has to be clearly stated for both figures. Consider adding extra information about the order into the Fig.2 legend.

Discussion section should be improved. Two suggestions here. First, the authors show that the drugs blocking HSPs are inhibiting HAdV growth. Therefore it is important to discuss what is known about the HSPs in HAdv infection and even speculate how the potential anti-HSP drugs act mechanistically on HAdV growth. Second, similar comment applies to CRM1 inhibitors as CRM1 functions have been well studied in HAdV infection. Hence, the authors viewpoints how the drugs against CRM1 will mechanistically act on HAdV growth would be appreciated.
